# Understanding COVID-19 Epidemiology and Implications for Control: The Experience from a Greek Semi-Closed Community

**DOI:** 10.3390/jcm10132765

**Published:** 2021-06-23

**Authors:** Ourania S. Kotsiou, Dimitrios Papagiannis, Evangelos C. Fradelos, Garyfallia Perlepe, Angeliki Miziou, Dimitra S. Siachpazidou, Konstantinos I. Gourgoulianis

**Affiliations:** 1Faculty of Nursing, School of Health Sciences, University of Thessaly, GAIOPOLIS, 41110 Larissa, Greece; efradelos@uth.gr; 2Department of Respiratory Medicine, Faculty of Medicine, School of Health Sciences, University of Thessaly, BIOPOLIS, 41110 Larissa, Greece; perlepef19@gmail.com (G.P.); kellymiz95@yahoo.com (A.M.); sidimi@windowslive.com (D.S.S.); kgourg@med.uth.gr (K.I.G.); 3Public Health & Vaccines Lab, Department of Nursing, School of Health Sciences, University of Thessaly, GAIOPOLIS, 41110 Larissa, Greece; dpapajon@uth.gr

**Keywords:** coronavirus, seroprevalence, semi-closed community

## Abstract

Background: We aimed to estimate the SARS-CoV-2 antigen and antibody seroprevalence in one of the worst-affected by the pandemic semi-closed communities in Greece, Deskati, and evaluate the sociodemographic and clinical correlations of functional antibody responses. Methods: The Ag2019n-CoV V1310/V1330 Rapid Test (Prognosis Biotech, Greece) was used for antigen detection. The Rapid Test 2019-nCoV Total Ig, V1210/V1230 (Prognosis Biotech, Greece), and the SARS-CoV-2 IgG II Quant method (Architect, Abbott, Illinois, USA) were used for antibody testing. Results: None of the participants had a positive antigen result. SARS-CoV-2 seropositivity ranged from 13% to 45% in the study population, depending on the method. One-third of the participants with known past infection had a positive antibody test result 77 ± 13 days after infection. Two-fifths of infections determined by serology were asymptomatic. The advancing age and hospitalization predicted seropositivity among patients with past infection. Half of the participants who tested positive for antibodies were not aware of past infection. Conclusions: High-burden contexts in Greece, such as Deskati, are not so far from herd immunity thresholds. We highlighted the value of low-cost serosurveys targeting both symptomatic and asymptomatic populations to evaluate the natural immune response to SARS-CoV-2 in nonvaccinated susceptibles and design evidence-based policies for lifting lockdowns.

## 1. Introduction

The first case of new coronavirus disease (COVID-19), caused by the novel severe acute respiratory syndrome coronavirus 2 (SARS-CoV-2), was documented in the province of Wuhan, China, in December 2019 [1]. Following the spread of the pandemic across the globe, the first COVID-19 case in Greece was reported on 26 February 2020 [2]. The Greek Centre for Disease Prevention and Control (CDC) on 13 March 2020 announced 190 confirmed cases and the first death from the novel coronavirus [2]. During the first wave of the pandemic, Greece implemented very harsh preventive measures to delay the virus spread. From 10 March to 4 May, wide strict restrictions on public movements were announced, and essential stores, restaurants, and bars were restricted to takeaway service only, thus achieving a low COVID-19 disease burden [3,4].

According to the Greek CDC data, the second wave of the pandemic began early in October 2020 and dramatically rose during November 2020 with high morbidity and mortality rates [5]. Some regions where incidence rates tended to be low during the first wave of the pandemic showed dramatically high mortality and morbidity rates during the second wave [6]. The same epidemic curves of the COVID-19 outbreak have also been recorded in other places worldwide [7,8].

Infectious diseases present particular challenges to semi-closed communities [9]. Many circumstances govern the spread of infection in a semi-closed community. On the one hand, an epidemic could be avoided if certain conditions were fulfilled. Otherwise, it could spread rapidly in humans as there was no existing immunity [9]. The study of such communities may lead an observer to identify the history and potential modes of transmission and effectively evaluate the impact of infection control strategies [4,9]. In other words, the occurrence of an epidemic in a semi-closed community offers the opportunity for a thorough investigation and is, therefore, of more than usual interest [9].

Deskati is a mountain town built at an altitude of 860 meters above sea level, with a regional population size of around 3508 people, constituting one of the largest semi-closed communities in central Greece. It is also one of the most characteristic remote indigenous communities due to limited access to medical care. Therefore, the recent coronavirus outbreak history in Deskati has been of great national and epidemiologic interest [10].

The first COVID-19 case in the municipality of Deskati was confirmed on 8 October 2020 [10]. Two more cases were reported on 25 October 2020 and one more case on 31 October 2020 [10]. The Greek National Holiday of 28 October interfered with the continuity of events, given that the municipality of Deskati welcomes visitors. During this holiday period, a number of carriers spread the causative organism resulting in a greater than that which existed at the close of the previous month [10]. The household and outdoor gatherings potentially contributed to community spread and transmission [10]. Indeed, 33 infected people out of 46 (71.7%) who had been tested on 5 November 2020 and 36 infected persons out of 138 (26%) who had been tested on 6 November 2020 were reported, respectively [10]. As of 23 November, 182 locally confirmed COVID-19 cases were reported by the National Organization of Public Health (EODY), although secondary schools had already been closed and restriction measures against non-essential movement had been announced since 5 November 2020. Notably, the rest of the nation implemented infection preventive and control regulations relating to free movement and business activity later, from 7 November 2020 [10]. The occurrence of cases, lockdown initiation, and significant time points in the course of the pandemic in Deskati are presented in Figure 1.

Unfortunately, 62 people died due to coronavirus during the one-month period from November to December 2020 [10]. Meanwhile, the number of hospitalizations and/or intensive care unit stays was not precisely known, estimated to approach one-third of infected patients [10]. A decline of incidence was noted in December 2020, with 13 confirmed COVID-19 cases, while none tested positive in January 2021; this was strongly linked to tougher lockdown rules that had been introduced earlier [10].

The close epidemiological monitoring of this closed population shed light on the parameters that were responsible for some of the features of the rise and fall of incidence of infection during the second wave. Nevertheless, the antibody seroprevalence is not known in this population. For the first time, we aimed to estimate the SARS-CoV-2 antigen and antibody seroprevalence and in-depth evaluate the sociodemographic and clinical correlations of functional antibody responses in one of the worst-affected by the second pandemic wave semi-closed communities in Greece.

## 2. Materials and Methods

A surveillance program was conducted in the municipality of Deskati on 29 January 2021. All the residents of Deskati were invited to participate in the study by the local authority and had been notified of the time and place thereof. Furthermore, we recruited participants by announcing the research on media while local officials organized a one-month recruitment campaign. Thus, every member of the larger population of Deskati had an equal chance of being included in the sample. There were no exclusion criteria. Importantly, residents did not differ in the exposure history due to the limited number of entrances and human mobility.

The Ethics Committee of the University Hospital of Larissa approved the study, and all subjects provided written informed consent. Following consent, demographic information, and data regarding past PCR-confirmed COVID-19 infection documented in the medical records, hospitalization and/or home isolation due to coronavirus, previous SARS-CoV-2 antibody testing, patient contacts, the participants’ travel history, and respiratory symptoms the last four months, medical, and smoking history were recorded on questionnaire forms for all participants. 

The Rapid Test Ag 2019n-CoV, V1310/V1330 (Prognosis Biotech, Larissa, Greece), was used for the qualitative detection of the nucleocapsid protein antigen from SARS-CoV-2 in human nasopharyngeal swab specimens [11], utilizing a lateral flow-based technology and providing 96.33% (95% CI 90.87–98.99%) sensitivity and 99.62% (95% CI 97.88–99.99%) specificity (Appendix A), and no cross-reaction with other viruses or other limitations. A recent meta-analysis showed that the sensitivity of rapid antigen tests for SARS-CoV-2 infection is generally much lower than what manufacturers reported [12]. According to the World Health Organization (WHO) recommendation, the cases with a positive rapid test or inconclusive results were further tested by a real-time reverse transcription–polymerase chain reaction (RT-PCR) molecular diagnosis for SARS-CoV-2, using a TaqPath™ COVID-19 CE-IVD RT-PCR kit (Thermo Fisher Scientific Inc., Waltham, MA, USA), detecting the ORF1ab, S, N genetic loci of SARS-CoV-2 with an analytical sensitivity and specificity of greater than 95% [13,14]. RT-PCR testing has been available at no cost nationwide at secondary and tertiary hospitals in Greece.

A low-cost double antigen sandwich lateral flow immunoassay that can simultaneously and rapidly detect IgG, IgM, and IgA directed against the S1 subunit of the spike protein of SARS-CoV-2 (Rapid Test 2019-nCoV Total Ig, V1210/V1230, Prognosis Biotech, Larissa, Greece) was used at a large scale for the qualitative determination of COVID-19 antibodies in the blood specimens that were conveniently obtained by pricking the finger with a triangular head and a sharp tip. This method’s clinical diagnostic sensitivity and specificity have been 98.75% (95% CI: 0.9325 to 0.9978), and 100% (95% CI: 0.9674 to 1), respectively [15].

In addition, the SARS-CoV-2 IgG II Quant ELISA method (Architect, Abbott, IL, USA), a chemiluminescent microparticle immunoassay, was used for the qualitative and quantitative determination of IgG antibodies against the spike receptor-binding domain (RBD) of SARS-CoV-2 in serum specimens from a smaller randomly selected element of the study population [16,17]. We used a table of random numbers in excel to select each member of the sample set for ELISA testing. By using a random number table, all members in the study population had an equal and independent chance of being selected for the sample sub-group. This ELISA method has a sensitivity of 99.9% and specificity and 100% for detecting the IgG antibody. Experienced doctors and one experienced biologist who had specialized theoretical and skills training on operating standards of nasopharyngeal swab sampling and self-protection conducted all the SARS-CoV-2 antigen and antibody tests [16,17].

The chi-square test was used to make comparisons between frequencies. Unpaired *t*-test was used for comparing parametric data between two groups, while non-parametric data were analyzed with the Mann–Whitney U test. Parametric data comparing three or more groups were analyzed with one-way ANOVA and Tukey’s multiple comparisons test, while non-parametric were analyzed with the Kruskal–Wallis test and Dunn’s multiple comparison test. Spearman’s correlation was used for correlation analysis. Multiple logistic regression was used to examine a series of predictor variables to determine those that best predict a positive SARS-CoV-2 antibody test result. Statistical analyses were performed with IBM SPSS Statistics for Windows, version 23.0, Armonk, NY: IBM Corp.

## 3. Results

In total, 388 volunteers (180 males, 208 females) participated in our study. The mean age of the population was 51.5 ± 18.6 (min: 9, max: 92 years). All participants filled out the survey and approved biological specimen collections. The sociodemographic, clinical characteristics, and the history of past COVID-19 disease or exposure for all participants are presented in Table 1.

In total, 26.3% of the study population had confirmed nucleic acid amplification tests (NAATs)-confirmed COVID-19 or tested positive for antibodies. None of the study participants had a positive or inconclusive SARS-CoV-2 rapid antigen result at that point in time. As far as the antibody detection is concerned, we found a SARS-CoV-2 antibody prevalence of 13% in the whole study population (51/388 participants) using rapid lateral flow immunoassays, and a prevalence of 45% (31/69 participants) by using ELISA tests in 69 randomly selected individuals from the whole study population; 14 of the 69 participants (20%) had a prior positive RT-PCR. Given that study results based on random samples are considered generalizable, we conclude that the SARS-CoV-2 antibody prevalence ranged from 13% to 45% in the study population, depending on the method used (lateral flow immunoassays or ELISA, respectively). More specifically, we found a lower sensitivity of the rapid antibody test compared to the ELISA. The lateral flow immunoassay showed a sensitivity of 45%, a specificity of 100%, and an accuracy of 75%.

Taking into account that the combination of results of multiple antibody diagnostic methods can improve the diagnostic accuracy [18,19], all people with a positive result from either one of the two antibody tests were assumed to have a past COVID-19 infection. Therefore, among all participants, a total of 69 participants (17.8%) were tested positive for SARS-CoV-2 antibodies either with one or both serological tests.

Of the positive group, 15% (10/69) had been already testing antibody positive, although one-fifth (15/69) got tested with another antibody test the previous 52 ± 47 days. One-third (35/102) of the participants who knew that they had been infected were found with a positive antibody test result in our study. The mean time has passed since the confirmed diagnosis was 77 ± 13 days.

Of the 102 patients, 67 (65.7%) who were previously tested positive PCR had a negative antibody test in our study. As we have previously mentioned, the mean time interval between the positive PCR and the negative antibody test was 77 ± 13 days. The characteristics of the patients with previously confirmed SARS-CoV-2 infection who tested positive in comparison with those tested negatives for virus antibodies are presented in Table 2.

According to this data, among those being previously infected with SARS-CoV-2, 25.8% reported having an asymptomatic disease. A multiple logistic regression model was conducted considering the antibody rapid or ELISA test result (positive vs. negative) as a dependent variable in patients with previously confirmed SARS-CoV-2 infection (Table 3). The analysis showed that the age (OR 1.05; 95% CI 1.01–1.09; *p* = 0.020) and the past COVID-19 hospitalization (OR 3.79; 95% CI 1.10–12.9; *p* = 0.034) were independent predictors for SARS-CoV-2 seropositivity.

No significant differences in the likelihood of having SARS-CoV-2 positive antibodies were found for gender, body mass index (BMI), smoking status, the presence of comorbidities, immunosuppression, or symptomatic disease in patients with a past confirmed SARS-CoV-2 disease at univariate analyses; thus, they were not used as independent variables in the regression analysis.

Interestingly, 49% (34/69) of the participants diagnosed with positive antibody results were not aware of the past infection and did not suspect having ever been infected. Overall, two-fifths (41%) of infections determined by serology were asymptomatic. Alternatively, among those who reported being negative for past infection, 11.9% (34/289) were tested positive for antibodies that were reflective of a past infection (34/388, 8.8% of the total study population).

The characteristics of the patients who tested positive and had previously confirmed SARS-CoV-2 infection compared to those diagnosed positive and were not aware of the past infection are presented in Table 4.

Those individuals who unexpectedly tested positive were younger and had lower BMI, fewer comorbidities and medication use, respiratory symptoms contributing to another disease, health care visits for any reason, and no hospitalizations compared to those patients with known past infection who tested positive for antibodies.

## 4. Discussion

This is the first study conducted in one of the most hard-hit semi-closed communities in Greece to estimate the SARS-CoV-2 antigen and antibody seroprevalence almost four months (January) after the initiation of the pandemic wave in the area (October). No active cases were found in the study population at this time point. We found that SARS-CoV-2 seropositivity ranged from 13% to 45% in the study population, depending on the antibody testing method. One-third (35/102) of the participants with known past infection had a positive antibody test result in our study. The mean time has passed since the confirmed diagnosis was 77 ± 13 days for those with known past infection. An important finding of this study was that the advancing age and the past COVID-19 hospitalization were positively associated with SARS-CoV-2 seropositivity and could independently predict a positive antibody result among patients with previously confirmed SARS-CoV-2 infection. Overall, two-fifths (41%) of infections determined by serology were asymptomatic. It was somewhat surprising that almost half of the participants who tested positive for antibodies were not aware of past infection and did not suspect having ever been infected. These individuals were younger and less obese and had fewer comorbidities, medication use, respiratory symptoms, health care visits for any reason, and no hospitalizations compared to people known to have COVID-19 who tested positive for antibodies.

Τhe duration of rises in titers of antibodies is currently unknown, and there is very little data beyond 35 days post-symptom onset [20]. Against weak and contradictory evidence in the literature, seroprevalence studies are central to provide a direct estimate of the percentage of individuals being immune to the virus and the longevity of the immune response in people with previously confirmed COVID-19 disease [21]. Furthermore, given that antibodies are a reliable measure of prior infection, the seroprevalence acts as a proxy for the cumulative incidence of COVID-19 until that time point after infection. Τo achieve accurate estimates from seroprevalence studies, it is important to recruit a representative sample of the population of interest [20]. Our sample is representative of the overall population of Deskati to make conclusions, as 11% of the targeted population were tested without excluding certain population members, exceeding the minimum sample size needed to estimate the population with a Confidence Level of 95%.

Moreover, to overcome seropositivity misclassification, we used two independent immunological tests with high sensitivity and specificity [18,19,21]. A recent metanalysis showed that lateral flow and ELISA-based methods perform better in terms of sensitivity (90–94%), which could be a safer choice at this stage of the pandemic [22]. We found that SARS-CoV-2 seropositivity ranged from 13% to 45% in the study population, depending on the assay used. The wide variance in seroprevalence estimates put the pros and cons of the two methods closely parallel. Higher seropositivity was detected using the Abbott SARS-CoV-2 IgG assay in the randomly selected sample of the population. Although this method is both precise and accurate, it is also more costly and technically demanding, requiring invasive blood sampling and subsequent centrifugation to collect a minimum of 100 μL of serum or plasma, compared to the rapid antibody test [16]. On the other hand, rapid antibody testing can provide a bigger window for mass detections and a practical approach for continuous surveillance due to the low cost, high speed of detection, and its minimally invasive character requiring a single drop of blood from a finger-stick puncture. Moreover, a combined IgA/IgG/IgM test has been reported to be a better choice in terms of sensitivity than measuring either antibody alone, showing the raises of all three main isotypes. However, the primary aim of this study was not to perform a head-to-head comparison between the two serodiagnostic assays, but to combine those diagnostic test results to increase the overall accuracy [18,19]. We conclude that the combination of those two tests was highly effective for identifying subjects who have antibodies to COVID-19 at a low cost. More importantly, the health benefits of this strategy exceed any cost.

For seroprotection estimates not to be biased [21], individuals were enrolled at a late phase of the second epidemic wave so as to be more likely to be seropositive, painting a clear picture of the immunoprotection in the targeted population. Serological findings over a longer period than 30 days since post-symptom onset remain limited and conflicting. In that context, it has been supported that the sensitivity of antibody tests is too low in the first week since symptom onset to have a primary role in diagnosing COVID-19 [20]. However, they may still have a role complementing another testing in individuals presenting later, when RT-PCR tests are negative or are not done, especially if they are used 15 or more days after the onset of symptoms [20]. Conversely, other studies support that anti-SARS-CoV-2 antibody responses follow a rapid increase within the first three weeks after symptoms and reduce subsequently [23]. However, the ability to detect anti-SARS-CoV-2 IgG antibodies remained robust for up to 6 months in a large proportion of previously virus-positive screened patients [23]. We found that one-third of the subjects who knew that they had been infected had a positive antibody test result 77 ± 13 days after the confirmed diagnosis, a finding that triggers further and longitudinal analysis of protective immunity to SARS-CoV-2 in order to answer the question of long-lasting protective immunity against SARS-CoV-2.

The SARS-CoV-2 seroprevalence in the general population worldwide varied from 0.37% to 22.1%, with a pooled estimate of 3.38% (95% CI 3.05–3.72%). On a regional level, seroprevalence varied from 1.45% (0.95–1.94%, South America) to 5.27% (3.97–6.57%, Northern Europe), related to the serological assay used [24]. Although a considerable amount of literature has been published on the seroprevalence of COVID-19 in specific population samples such as hospitalized patients, blood donors, healthcare workers [25], pregnant women, etc. [26], there is very little published research on seropositivity outside of clinical settings, in semi-closed or closed communities such as ethnic or religious minorities. Limiting the study to groups with high infection rates and risk of exposure can mitigate bias while improving power. Preliminary results from a serosurvey in Mumbai, India, showed 58% of slum-dwellers versus 17% of the non-slum population had antibodies to SARS-CoV-2 [24,26,27]. The semi-closed community of Deskati has among the highest SARS-CoV-2 seroprevalence rate described in the literature. At that time, high-burden contexts in Greece, such as Deskati, were not so far from herd immunity thresholds, given that basic models for COVID-19 suggest that herd immunity is achieved when 45–75% of people are immune.

Havers et al. reported that it is likely that greater than 10 times more SARS-CoV-2 infections occurred for most sites than the number of reported COVID-19 cases [28]. Importantly, our findings indicating that the estimated prevalence of SARS-CoV-2 antibodies was likely more widespread than indicated by the number of reported SARS-CoV-2 cases. Notably, one-fourth of the study population reported a known infection; however, an additional 11.9% among the participants with previously unknown infection tested positive for antibodies. These results corroborate the findings of a great deal of the previous work in that context. A study by Bruckner et al., supported that seroprevalence among adults of Orange County, the sixth-largest county in the US with a population of 3.2 million, is seven-fold greater than that using official county statistics [29]. In the present study, we found that individuals who first diagnosed with past SARS-CoV-2 infection by serology were younger and less obese and had fewer comorbidities, medication use, respiratory symptoms which were attributed to another disease, health care visits for any reason, and no hospitalizations compared to positives with known past COVID-19 infection. Host factors including age and comorbid conditions are key determinants of disease severity and progression [30]. Age-related decline, and immunosenescence and inflammaging play a major role in heightened vulnerability to severe COVID-19 outcomes in older adults [30]. Conversely, younger people would be less likely to have severe disease, more likely to be asymptomatic or have mild symptoms, and thus less likely to be tested [30,31]. It has been reported that possible reasons for milder presentations in children and adolescents include frequent contact with seasonal coronaviruses, the presence of cross-reactive antibodies, and/or co-clearance with other viruses [32]. However, in young adults, obesity and metabolic syndrome have been associated with adverse outcomes [33]. Recently, it has been supported that asymptomatic infection can induce the same humoral immunity as non-severe COVID-19 in young adults [34].

Overall, in our study, two-fifths (41%) of infections determined by serology were asymptomatic. Exploring data derived from MERS-CoV cases, it has been reported that some patients with mild symptoms of the disease may fail to develop detectable levels of antibodies [35]. Serological findings in patients with non-severe COVID-19 are scarce and conflicting. Even though our findings indicate that several patients with both mild, severe, or no symptoms could have positive antibody concentrations, at the same time, not all patients develop detectable levels of antibodies in these assays, despite the fact that 77 days or more had passed since the disease onset. An anticipated result was the serum-antibody responses to SARS-CoV-2 were differentiated by the clinical severity of disease and age among patients with a past confirmed infection. The discrepancies between the results may partly be explained by different target antigens used in antibody detection in various studies and short follow-up (less than 25–50 days).

The past COVID-19 hospitalization was positively associated with SARS-CoV-2 seropositivity and could independently predict a positive antibody result among patients with previously confirmed SARS-CoV-2 infection. Our findings accord with earlier observations that failed to detect antibodies in patients with mild disease showing that patients with severe COVID-19 seroconvert earlier and develop higher concentrations of natural SARS-CoV-2-specific IgG than patients with mild symptoms [36]. Another study with 34 hospitalized patients with COVID-19 presented increased IgG levels until five weeks of the disease onset, followed by consistent levels up to 7 weeks [36]. Vogelzang et al. supported that in nonhospitalized patients, the antibody response is weaker but follows similar kinetics, as observed in hospitalized patients [37]. Conversely, Long et al. reported that 97% of 37 patients with mild COVID-19 had decreased levels of IgG two to three months post-symptom onset [38]. Kong et al. suggested that early seroconversion and high antibody titer were linked with less severe clinical symptoms [39].

Besides, we have shown that another independent predictor for SARS-COV-2 antibody positivity among previously confirmed patients was the increasing age. It is widely reported that children have milder disease compared with adults [40]. On the other hand, the significantly higher mortality rates seen in the elderly compared with young children are likely to be driven in part by an impaired immune response in older individuals [40]. However, very little research has been carried out on the effects of aging in SARS-COV-2 natural antibody response, the kinetics of viral clearance, and antibody production across age groups. In accordance with the present results, Klein et al. very recently documented that male sex, advancing age, and hospitalization for COVID-19 were associated with greater natural antibody and IgG responses to SARS-CoV-2, across the serological assays, among 126 potential convalescent plasma donors [40]. Zang et al. supported that higher IgG titers were correlated with worse COVID-19 outcomes, which is also reflected in the link between greater antibody titers and older age [41]. Nevertheless, no sex-related differences in antibody responses to SARS-CoV-2 were detected in our study.

One major limitation of this study was that we used a random sample rather than the entire population to examine seropositivity with the SARS-CoV-2 IgG II Quant ELISA detection method, given that this strategy was time- and cost-efficient. Nevertheless, to overcome seropositivity misclassification, we used two independent immunological tests with high sensitivity and specificity. Moreover, a table of random numbers was used to determine which participants were to be selected. Another limitation is that seropositivity at the population level might imperfectly represent cumulative incidence due to possible changes over the course of the antibody response. For seroprotection estimates not to be biased, individuals were enrolled at a late phase of the second epidemic wave so as to be more likely to be seropositive, painting a clear picture of the immunoprotection in the targeted population. Moreover, the study population is limited to one geographic area, translated into a lack of generalizability. Participants and non-participants did not differ in the exposure history due to the limited number of entrances and human mobility. Although the study population was randomly selected, we cannot exclude that participants might have characteristics associated with willingness to participate. In spite of its limitations, the key strengths of the study were that we recruited participants in a way that did not systematically favor those with unusually high or low levels of exposure to reduce volunteer bias, and we considered demographics and other information about participants to facilitate adjustment of results.

## 5. Conclusions

The major contribution of this study is that for the first time, we provide data regarding the SARS-CoV-2 antigen and antibody seroprevalence in one of the worst-affected by the second pandemic wave semi-closed communities in Greece at a four-month time point after the initiation of the pandemic wave in the area. At the same time, we present an in-depth evaluation of sociodemographic and clinical correlates of functional antibody responses. We found that SARS-CoV-2 seropositivity ranged from 13% to 45% in the study population of Deskati, depending on the antibody testing method. One-third of the participants with known past infection were found with a positive antibody test result after a mean duration of 77 ± 13 days since infection. Overall, in our study, two-fifths of infections determined by serology were asymptomatic, which highlights the value of programs targeting both symptomatic and asymptomatic populations. The advancing age and the past COVID-19 hospitalization were independent predictors of a positive antibody result among patients with previously confirmed SARS-CoV-2 infection. Almost half of the participants who tested positive for antibodies were not aware of past infection and did not suspect having ever been infected. These individuals were younger and less obese and had fewer comorbidities, medication use, respiratory symptoms, health care visits for any reason, and no hospitalizations compared to people known to have COVID-19 who tested positive for antibodies. This is a fast-moving field, and the population of Deskati is deemed to be of great epidemiological interest. We aimed to record the traces of a past active circulation of the virus in this fragile population again during the spring, identify the kinetics of virus antibody responses at later stages of the disease, and evaluate the immunization status and the seroprevalence in vaccinated individuals in this semi-closed community.

## Figures and Tables

**Figure 1 jcm-10-02765-f001:**
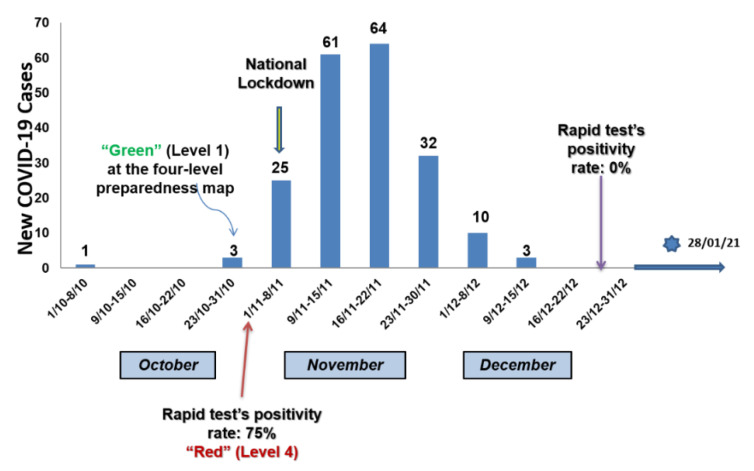
The occurrence of cases, lockdown initiation, and significant time points in the course of the pandemic in Deskati.

**Table 1 jcm-10-02765-t001:** Characteristics of the study population stratified by gender, *n* = 388.

Variable	Total (*n* = 388)	Males (*n* = 180)	Females (*n* = 208)	
Age (years)	51.5 ± 18.6	52.1 ± 19.3	51.1 ± 18.1	
BMI (mg/kg^2^)	32 ± 6	32 ± 5	32 ± 6	
Smoking status				
Ex-smokers *n*, (%)	73 (18.8)	58 (32.2)	15 (7.2)	
Current smokers *n*, (%)	74 (19.1)	45 (25)	29 (13.9)	
Non-smokers *n*, (%)	241 (62.1)	77 (42.8)	164 (78.8)	
Comorbidities (yes, *n* (%)	211 (54.4)	101 (56.1)	110 (52.9)	
Medication (yes), *n* (%)	187 (48.1)	91 (50.6)	96 (46.2)	
Immunosuppression (cancer, autoimmune disease) (yes), *n* (%)	7 (1.8)	3 (1.7)	4 (1.9)	
Respiratory symptoms the last four months (yes), *n* (%)	164 (42.3)	69 (38.3)	95 (45.7)	
Health care visit the last 4 months (yes), *n* (%)	81 (20.9)	40 (22.2)	41 (19.7)	
Travels the last 4 months, (yes), *n* (%)	58 (14.9)	29 (16.1)	29 (13.9)	
Previous exposure to a COVID-19 positive case, *n* (%)	20 (5.2)	10 (5.6)	10 (4.8)	
Past PCR-confirmed COVID-19 infection, (yes), *n* (%)	102 (26.3)	51 (28.3)	51 (24.5)	
Past COVID-19 hospitalizations, (yes), *n* (%)	17 (4.4)	11 (6.1)	6 (2.9)	
Mean duration of hospitalization (days)	8 ± 5	9 ± 5	9 ± 7	
Mean duration of home isolation (days)	21 ± 11	20 ± 10	21 ± 12	
Previous SARS-CoV-2 antibody testing, (yes), *n* (%)	62 (16%)	29 (16.1)	33 (15.9)	0.528

Data are expressed as mean ± SD or as frequencies (percentages). Abbreviations: BMI, body mass index; COVID-19, coronavirus disease 2019; PCR, Polymerase Chain Reaction; SARS-CoV-2, severe acute respiratory syndrome coronavirus 2.

**Table 2 jcm-10-02765-t002:** Characteristics of the patients with previously confirmed SARS-CoV-2 infection who tested positive in comparison with those tested negatives for virus antibodies, *n* = 102.

Variable	*n* of Individuals with Previously Confirmed SARS-CoV-2 infection (*n* = 102)	Positive Antibody Test (*n* = 35)	Negative Antibody Test (67)	*p*-Value
Gender Male *n*, (%)	51 (0.5)	20 (57.1)	31 (46.3)	0.202
Age (years)	51.2 ± 18.6	61.0 ± 14.6	46.0 ± 18.4	<0.001
BMI (mg/kg^2^)	32 ± 6	33 ± 5	32 ± 6	0.385
Smoking status Ex- or non-smokers *n*, (%) Current smokers *n*, (%)	89 (87.3) 13 (12.7)	34 (97.1) 1 (2.9)	55 (82.1) 12 (17.9)	0.025
Comorbidities (yes), *n* (%)	64 (62.7)	25 (71.4)	39 (71.4)	0.136
Medication (yes), *n* (%)	53 (52.0)	25 (71.4)	28 (41.8)	0.004
Immunosuppression (cancer, autoimmune disease) (yes), *n* (%)	4 (3.9)	3 (8.6)	1 (1.5)	0.116
Respiratory symptoms the last four months (yes), *n* (%)	74 (72.5)	26 (74.2)	48 (71.6)	0.484
Health care visit the last 4 months (yes), *n* (%)	47 (46.1)	20 (57.1)	27 (40.3)	0.079
Past COVID-19 hospitalizations, (yes), *n* (%)	17 (16.6)	11 (31.4)	6 (8.9)	<0.001

Data are expressed as mean ± SD or as frequencies (percentages). Abbreviations: BMI, body mass index; COVID-19, coronavirus disease 2019; PCR, polymerase chain reaction; SARS-CoV-2, severe acute respiratory syndrome coronavirus 2.

**Table 3 jcm-10-02765-t003:** Multiple logistic regression model with antibody test result (positive vs. negative) as a dependent variable.

Variables	B	SE	Wald	Sig	Exp (B)	95% CI for Exp (B)	
						Lower	Upper
Age ^a^	0.05	0.02	6.16	0.01	1.05	1.01	1.09
Medication	0.12	0.61	0.04	0.85	1.12	0.34	3.69
Hospitalization	1.45	0.63	5.40	0.02	4.27	1.26	14.52

^a^ Continuous variable. Abbreviations: B, B coefficient; CI, confidence interval; Exp (B), exponentiation of the SE; standard error; sig, significance.

**Table 4 jcm-10-02765-t004:** The characteristics of the patients who tested positive and had previously confirmed SARS-CoV-2 infection compared to those diagnosed positive and were not aware of the past infection, *n* = 69.

Variable	*n* of Previously Confirmed SARS-CoV-2 Infection (*n* = 69)	Previously Known COVID-19 Disease (*n* = 35)	Previously Unknown COVID-19 Disease (34)	*p*-Value
Gender Male *n*, (%)	35 (50.7)	20 (57.1)	15 (44.1)	0.200
Age (years)	51.2 ± 18.6	61.0 ± 14.6	47.9 ± 18.2	0.002
BMI (mg/kg^2^)	32 ± 6	33 ± 5	30 ± 5	0.037
Smoking status Ex- or non-smokers *n*, (%) Current smokers *n*, (%)	60 (87.0) 9 (13.0)	34 (97.1) 1 (2.9)	26 (76.5) 8 (23.5)	0.012
Comorbidities (yes, *n* (%)	39 (56.5)	25 (71.4)	14 (41.2)	0.011
Medication (yes), *n* (%)	35 (50.7)	25 (71.4)	10 (29.4)	0.001
Immunosuppression (cancer, autoimmune disease) (yes), *n* (%)	3 (4.3)	3 (8.6)	0	0.116
Respiratory symptoms the last four months (yes), *n* (%)	40 (58.0)	26 (74.2)	14 (41.2)	0.005
Health care visit the last 4 months (yes), *n* (%)	23 (33.3)	20 (57.1)	3 (8.8)	<0.001
Past COVID-19 hospitalizations, (yes), *n* (%)	11 (15.9)	11 (31.4)	0	<0.001

Data are expressed as mean ± SD or as frequencies (percentages). Abbreviations: BMI, body mass index; COVID-19, coronavirus disease 2019; PCR, polymerase chain reaction; SARS-CoV-2, severe acute respiratory syndrome coronavirus 2.

## Data Availability

The data that support the findings of this study are available on request from the corresponding author, O.S.K.

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
