# Peer review of "Understanding COVID-19 Epidemiology and Implications for Control: The Experience from a Greek Semi-Closed Community"

_jcm, 2021, doi:10.3390/jcm10132765_

Round 1
Reviewer 1 Report
The study is well thought-out and carefully planned. Three major limitations: 1) the study population is limited to one geographic area and is lack of generalizability; 2)relative to the intended population to recruit, the number of participants is very low. 3) The study need to address if participants and non-participants have any systematic difference, especially if they differ in the exposure history, disease history, characteristics, etc. It is plausible that people with a known exposure history are more willing to participating in the study. If so, it could inflate the positive rates reported in the study.
Author Response
RESPONSE TO REVIEWER 1:
- The study is well thought-out and carefully planned.
RESPONSE: We sincerely thank you for your kind words about our paper. We are delighted to receive positive feedback from you.
- Three major limitations: 1) the study population is limited to one geographic area and is lack of generalizability;
RESPONSE: Thank you for this comment. We agree with you that this is an important limitation that have been pointed out in the revised manuscript.
- 2)relative to the intended population to recruit, the number of participants is very low.
RESPONSE: Deskati has a regional population size of around 3.508 people. 347 participants (10%) were the minimum recommended size of this survey according to power analysis with a confidence level of 95%. We randomly recruited 388 participants and got responses from everyone.
- 3) The study needs to address if participants and non-participants have any systematic difference, especially if they differ in the exposure history, disease history, characteristics, etc.
RESPONSE: Thank you for this comment. Deskati is one of the most characteristic remote indigenous communities due to limited access to medical care in Greece. The first COVID-19 case in the municipality of Deskati was confirmed on the 8th of October 2020, and after that 71.7% of those who had been tested were positive. Participants and non-participants did not differ in the exposure history due to the limited number of entrances and human mobility. Although the study population was randomly selected, we cannot exclude that participants might have characteristics associated with willingness to participate. We agree with you that this is an important limitation that has been mentioned in the revised manuscript.
- It is plausible that people with a known exposure history are more willing to participating in the study. If so, it could inflate the positive rates reported in the study.
RESPONSE: Thank you for this remark. Only 5.2% of the participants reported a known previous exposure to a COVID-19 positive case and only 26.3% of the participants had a known past PCR-confirmed COVID-19 infection. Generally, we consider that volunteers did not disproportionately possess certain traits which affect the outcome.
We appreciate you taking the time to offer us your insights related to the paper. We found your feedback very constructive. We tried to be responsive to your concerns.
Reviewer 2 Report
This manuscript reports on SARS-CoV2 antigen and antibody testing results in participants recruited from a semi-closed community in Greece. While the results are interesting, more clarifications are needed with regards to participant selection. Furthermore, while the authors tested all study participants using a rapid antibody test, only few were tested using an ELISA-assay. The difference in the prevalence of antibodies between the two tests suggests a difference in sensitivity which is not discussed by the authors in sufficient detail. Also, some sections would require a review of the English language used to improve clarity.
Introduction
- Line 40: define what “table service” means or rephrase
- Lines 47-52: this section is difficult to understand, consider rephrasing to improve clarity. Potentially explain why the introduction of an infection in a closed community is different than the introduction in an open population
- Consider presenting the information relating the occurrence of cases in the form of an Epi-curve with arrows signalling important events (lockdowns, etc.). This graph can also be used to indicate the time points/ intervals when the study was conducted
Methods
- The authors describe the sensitivity of the antigen test as very high (~98%). Can the authors specify in which population the test was originally evaluated? According to systematic reviews of antigen tests, sensitivity is much lower (see for eg. Dinnes et al. doi: 10.1002/14651858.CD013705.pub2). The link referring to the source data (reference 11) is for the manufacturer website and it was directing to an incorrect page.
- Line 114 – it appears that something is missing after “triangular..”
- Line 116 and reference 14: the link for the reference is directing to an incorrect page. Please provide a reference for an independent evaluation of the test (other than the manufacturer).
- Line 123-124. Consider removing the positive and negative predictive values as they are prevalence-dependent.
- Provide more information on how the training for sample collection was conducted and clarify what types of swabs and testing were used for the antigen test (anterior nasal, nasopharyngeal, etc.)
- How available was PCR testing to the community? How was the past PCR result ascertained – was it documented in the medical records or by self-report?
- The authors should clarify how participant selection for recruitment into the study was done (were there any inclusion or exclusion criteria used, was it an active recruitment, eg. Door-to-door or participants had to come to the research team). Also please clarify how the random sampling was done to select the samples for testing.
- Line 134: provide the manufacturer, city country for SPSS
Results:
- Table 1: the p-values for the table can be omitted as the role of the table is to give an overview of the recruited participants
- Line 154: the authors should consider that there are several possibilities for the difference in sensitivity between the two antibody tests: for example, it can be due to a lower sensitivity of the rapid test vs. the ELISA, or it can be due to the selection of samples for ELISA testing not being truly random (please provide more detail on the selection process in the methods section).
- Line 161: how many participants with a prior positive PCR had a negative antibody test? For these – what was the time interval between the positive PCR/ onset of symptoms and the negative antibody test?
- Table 3: please clarify how many of the participants with previous infection included in the analysis had an ELISA result. Assuming that the selection for ELISA was random and participants who only had a rapid test done were also included in the logistic regression, then it is likely that participants were incorrectly classified due to the lack of sensitivity of the rapid test.
Discussion
- The authors should discuss the differences in accessing care/ testing for younger patients – they would be less likely to have severe disease, more likely to be asymptomatic or have mild symptoms and thus less likely to be tested
- Line 228: this statement may be outdated
- Line 236: more information on the selection process is needed preferably in the methods section to support the statement that the population is representative.
- Line 240: sensitivity might be an issue with the rapid test considering that all patients were tested by RDT and a subset by ELISA
Author Response
RESPONSE TO REVIEWER 2:
- This manuscript reports on SARS-CoV2 antigen and antibody testing results in participants recruited from a semi-closed community in Greece. While the results are interesting, more clarifications are needed with regards to participant selection.
RESPONSE: We sincerely thank you for your kind words about our paper. We are delighted to receive positive feedback from you. In the following pages are our point-by-point responses to each of your comments.
- Furthermore, while the authors tested all study participants using a rapid antibody test, only few were tested using an ELISA-assay. The difference in the prevalence of antibodies between the two tests suggests a difference in sensitivity which is not discussed by the authors in sufficient detail.
RESPONSE: Thank you for this comment. In the revision, we discussed this issue on page 5, lines 184 -186.
- Also, some sections would require a review of the English language used to improve clarity.
RESPONSE: We have gone through the whole manuscript and all grammatical mistakes and misspellings were corrected, as suggested.
- Introduction Line 40: define what “table service” means or rephrase
RESPONSE: Thank you for this comment. We have rephrased this sentence, as suggested.
- Lines 47-52: this section is difficult to understand, consider rephrasing to improve clarity. Potentially explain why the introduction of an infection in a closed community is different than the introduction in an open population
RESPONSE: We appreciate your suggestion. In the revised manuscript, we have rephrased this section to be more comprehensible to readers (page 2, lines 48-54)
- Consider presenting the information relating the occurrence of cases in the form of an Epi-curve with arrows signalling important events (lockdowns, etc.). This graph can also be used to indicate the time points/ intervals when the study was conducted.
RESPONSE: Thank you for this comment. In the revision, we have had introduced Figure 1, presenting the information relating to the occurrence of cases, lockdown initiation, significant time points and when the study was conducted.
- Methods: The authors describe the sensitivity of the antigen test as very high (~98%). Can the authors specify in which population the test was originally evaluated?
RESPONSE: Thank you for this comment. In the revised manuscript, we have corrected the sensitivity and specificity of the antigen test according to the clinical evaluation study (yet unpublished data), which is provided as supplementary data. The population in which the test was originally evaluated was individuals with suspected or confirmed SARS-COV-2 infection who approached the Emergency Department of the University Hospital of Larissa, Thessaly, Greece.
- According to systematic reviews of antigen tests, sensitivity is much lower (see for eg. Dinnes et al. doi: 10.1002/14651858.CD013705.pub2).
RESPONSE: Thank you for this interesting remark. We included this information in the revised manuscript (pages 3, lines 119-121).
- The link referring to the source data (reference 11) is for the manufacturer website and it was directing to an incorrect page.
RESPONSE: Thank you for this comment. We apologize for the mistake. In the revision, we have provided the correct link.
- Line 114 – it appears that something is missing after “triangular..”
RESPONSE: Thank you for this point. In the revision, we have corrected the phrase.
- Line 116 and reference 14: the link for the reference is directing to an incorrect page. Please provide a reference for an independent evaluation of the test (other than the manufacturer).
RESPONSE: Thank you for this remark. In the revision, we have provided another reference for an independent evaluation of the test, as suggested.
- Line 123-124. Consider removing the positive and negative predictive values as they are prevalence-dependent.
RESPONSE: Thank you for this comment. We have removed positive and negative predictive values, as suggested.
- Provide more information on how the training for sample collection was conducted and clarify what types of swabs and testing were used for the antigen test (anterior nasal, nasopharyngeal, etc.)
RESPONSE: Thank you for this comment. We have provided more information on how the training for sample collection was conducted and clarify what types of swabs and testing were used for the antigen test.
- How available was PCR testing to the community? How was the past PCR result ascertained – was it documented in the medical records or by self-report?
RESPONSE: Thank you for this comment. We have discussed these points on page 3, lines 126-127 and line 109, respectively.
- The authors should clarify how participant selection for recruitment into the study was done (were there any inclusion or exclusion criteria used, was it an active recruitment, eg. Door-to-door or participants had to come to the research team).
RESPONSE: Thank you for this valuable comment. We clarified this issue on page 3, lines 100-106.
- Also please clarify how the random sampling was done to select the samples for testing.
RESPONSE: Thank you for this direction. We clarified this point on page 4, lines 141-146.
- Line 134: provide the manufacturer, city country for SPSS
RESPONSE: Thank you for this comment. In the revised manuscript, we have provided the manufacturer, city country for SPSS (page 4, line 161).
- Results: Table 1: the p-values for the table can be omitted as the role of the table is to give an overview of the recruited participants
RESPONSE: Thank you for this suggestion. The p-values in Table 1 have been omitted, as suggested.
- Line 154: the authors should consider that there are several possibilities for the difference in sensitivity between the two antibody tests: for example, it can be due to a lower sensitivity of the rapid test vs. the ELISA, or it can be due to the selection of samples for ELISA testing not being truly random (please provide more detail on the selection process in the methods section).
RESPONSE: Thank you for this remark. We discussed this issue on page 4, lines 142-146 and page 6, lines 184-186.
- Line 161: how many participants with a prior positive PCR had a negative antibody test? For these – what was the time interval between the positive PCR/ onset of symptoms and the negative antibody test?
RESPONSE: Thank you for this point. We provided this information on page 6, lines 196-198.
- Table 3: please clarify how many of the participants with previous infection included in the analysis had an ELISA result.
RESPONSE: 69 randomly selected individuals from the whole study population had an ELISA result. 14/69 (20%) had a prior positive PCR.
- Assuming that the selection for ELISA was random and participants who only had a rapid test done were also included in the logistic regression, then it is likely that participants were incorrectly classified due to the lack of sensitivity of the rapid test.
RESPONSE: Taking into account that the combination of results of multiple antibody diagnostic methods can improve the diagnostic accuracy, the multiple logistic regression model was conducted considering the antibody test result (positive vs. negative) from either rapid or ELISA methods as a dependent variable (page 7, line 210).
- Discussion The authors should discuss the differences in accessing care/ testing for younger patients – they would be less likely to have severe disease, more likely to be asymptomatic or have mild symptoms and thus less likely to be tested
RESPONSE: Thank you for this direction. We discussed this issue on pages 10-11, lines 342-353.
- Line 228: this statement may be outdated
RESPONSE: Thank you for this comment. We have removed this statement, as suggested.
- Line 236: more information on the selection process is needed preferably in the methods section to support the statement that the population is representative.
RESPONSE: Thank you for this comment. We consider that volunteers did not disproportionately possess certain traits which affect the outcome. We recruited participants in a way that did not systematically favor those with unusually high or low levels of exposure to reduce volunteer bias. We provided more information on the selection process on page 3, lines 100-106; page 9, lines 268-273; and page 11, lines 408 -412.
- Line 240: sensitivity might be an issue with the rapid test considering that all patients were tested by RDT and a subset by ELISA
RESPONSE: Using a table of random numbers is a tool for randomly assigning participants, ensuring that each member of the study population has an equal chance of being selected for the study. However, the fact that only a subset tested by ELISA has already been pointed out as a limitation of our study (page 11, lines 394-396).
We are very grateful for the effort you dedicated to reviewing our submission, as well as your favorable comments. We hope the revised version of our manuscript will satisfy your concerns.
Round 2
Reviewer 2 Report
Thank you for submitting a revised document. I have no further comments.